# Psychiatric Nurses’ Knowledge and Practice Barriers to Administering Long-Acting Injectable (LAI) Antipsychotics in Taiwan: A Mixed-Methods Study

**DOI:** 10.3390/healthcare11121670

**Published:** 2023-06-06

**Authors:** Yao-Yu Lin, Wen-Li Hou, Mei-Ling Lin

**Affiliations:** 1Department of Nursing, Tsaotun Psychiatric Center, Ministry of Health and Welfare, Nantou 542, Taiwan; yylin@ttpc.mohw.gov.tw; 2College of Nursing, Kaohsiung Medical University, Kaohsiung 807, Taiwan; wlhou422@gmail.com; 3Department of Medical Research, Kaohsiung Medical University Hospital, Kaohsiung 807, Taiwan; 4Department of Nursing, HungKuang University, Taichung 433, Taiwan

**Keywords:** long-acting antipsychotics, injection, psychiatric nurses

## Abstract

Long-acting antipsychotic injections require that psychiatric nurses choose the proper injection site and technique to avoid harming patients. This research conducted a mixed-method study to examine the long-acting injectable antipsychotic (LAI) knowledge, practice, and administrative barriers in a sample of 269 psychiatric nurses from 3 public psychiatric hospitals in Taiwan. Self-report questionnaires showed female nurses exhibiting higher scores and older nurses demonstrating more knowledge. The dorsogluteal (DG) site was the most widely used for injections, with 57.6% of nurses using the Z-track method. Qualitative data analysis was performed on 20 psychiatric nurses who chose the DG site as their preferred injection site. There were two key themes. The first was a gap between the nurses’ knowledge of LAI administration and their actual practice. The second needed more confidence and training in using the ventrogluteal injection site. These results highlight the need for continued education and training to improve LAI practice among psychiatric nurses.

## 1. Background

The challenge of medication noncompliance among psychiatric patients is a significant concern for mental health professionals, with rates of non-adherence among those with schizophrenia, depression, and bipolar disorder estimated to be 56%, 50%, and 44%, respectively [1]. To address this issue, long-acting injectable antipsychotics (LAI) were developed in 1966 [2] and have since become a well-established treatment option. LAIs are given using intramuscular injection and supplement oral medications to prevent relapses and stabilize symptoms [3,4]. These medications sustain therapeutic levels in the body for an extended period and significantly mitigate noncompliance risk [3]. Moreover, they can enhance patients’ subjective well-being and overall quality of life [4,5].

Intramuscular injections (IMIs) have been used in clinical practice for several decades and are widely used for administering various medications to patients [6]. Psychiatric nurses are critical in implementing these injections for patients and must consider appropriate sites and techniques that require trained staff to ensure safe and accurate administration [7]. In this context, psychiatric nurses are critical in implementing these injections for patients.

There are two long-acting injectable antipsychotics (LAI) in clinical use: first-generation antipsychotics (FGAs) and second-generation antipsychotics. FGAs have a stable release and are typically made with vegetable oil. They are usually injected into the gluteal muscles [8,9]. On the other hand, second-generation antipsychotics are either coated with a biodegradable polymer (e.g., risperidone) or made with particles that are insoluble in water (e.g., olanzapine pamoate) [8]. The administration technique for intramuscular injection (IMI) is a crucial aspect when using both generations of LAIs for treating patients with psychosis. Nurses are responsible for administering the IMIs and must consider the appropriate site and technique and use their psychiatric knowledge and skills to make informed decisions. Nurses play a vital role in the IMI process, and their practices can directly reflect the quality of care they provide.

There are two sites for LAI injection: the deltoid site and the gluteal site. Generally, first-generation LAI medicines traditionally have been injected into a patient’s gluteal area. However, there is evidence against using the dorsogluteal (DG) site because, compared with the ventrogluteal (VG) site, it contains more major nerve and vascular structures that could be damaged during injection [10,11,12]. In fact, using the DG site can lead to a sciatic nerve injury [13]; using the V or G technique to find the VG site is a favorable alternative [14,15], as it is farther away from the sciatic nerve [13]. Significantly, the G technique is recommended for a safe IM injection VG site that considers a person’s anatomical structure [14].

Some studies have examined how often nurses use the recommended ventrogluteal (VG) site for intramuscular injections (IMIs) [9,16,17,18]. However, research shows that the VG site is not necessarily the preferred injection site for psychiatric nurses. There are three reasons why some nurses may not choose to use the VG site: it is a small anatomical area that can be difficult to locate, they worry that using this site may hurt patients, and some did not receive sufficient education on administering IMIs [6,16,19]. In a study where nurses received training on the VG site, after four months, the proportion of nurses using it as the injection site increased from 7.4% to 34.6%, and the proportion using the unsafe dorsogluteal (DG) site decreased from 76.5% to 48.1% [17]. However, after a period of time, a survey comparing the same subjects considering the VG site for injection found that only 11.8% of Australian psychiatric nurses used the VG site [16], which fell to 9.8% six years later [9]. Despite the VG site being recommended for IMIs for years, many nurses still need to choose the DG site [9,16,17,18]. Providing nurse education and training can help increase nurses’ use of the safer VG site, but the benefits may only last for a while.

Moreover, IMI methods can also result in skin irritation or damage after the injection, especially for long-acting antipsychotics that are oil-based [20]. Complications include subcutaneous lumps or hardening, muscle granulomas, medication seepage at the injection site [21,22], fibrosis, abscess [21], swelling, edema, redness, tenderness [21,22], and even sciatic nerve injury [13,15]. To reduce these risks, nurses may consider using different injection techniques, such as the two-needle method (using one needle to draw the medication and another to inject it), the Z-track method, the air-bubble technique [23,24], the direct injection technique, adjusting the needle angle, injecting deeply (inserting the needle two-thirds of the way into the patient’s skin), and rotating the injection site. To minimize issues with LAIs at the injection site, researchers have recommended using the Z-track method, air-bubble technique, or both. The Z-track approach helps prevent the medication from leaking into other tissues, thus reducing tissue damage. Based on a 2006 educational intervention study conducted in Australia regarding the injection practices of psychiatric nurses, it was found that at that time, 29% of nurses utilized the Z-track method for long-acting injections (LAIs). However, when the nurses were surveyed again after a 6-year period, the proportion of nurses using the Z-track method had significantly decreased to only 6% [9,16]. This means that despite receiving education on injection techniques, nurses may not consistently continue to implement them over time. In addition to these techniques, it is also recommended that nurses not massage the injection site after administration.

Pain at the injection site may discourage patients from using LAIs [20,25]. However, some techniques, such as using a different needle before injection and injecting more slowly, have been shown to help reduce pain intensity [26,27]. Psychiatric nurses play a vital role in ensuring the proper administration of LAIs. They need to understand the medication, choose the correct injection site, and monitor patients for potential side effects [4]. Previous research found that only a few nurses relied on relevant guidelines or the nursing literature when administering IMIs [18]. Unfortunately, many psychiatric nurses may need more knowledge or confidence to use evidence-based practices in LAI administration.

This study aims to address the following research questions related to the administration of long-acting injections (LAIs) by psychiatric nurses in Taiwan: (1) What is the level of knowledge among psychiatric nurses regarding the administration of LAIs? (2) What are the commonly used injection sites for administering LAIs among psychiatric nurses in Taiwan, and what are the reasons behind their choice? (3) What are the barriers that nurses encounter when selecting the ventrogluteal (VG) injection site for LAIs? By investigating these questions, this study seeks to gain insights into the knowledge, practices, and challenges faced by psychiatric nurses in administering LAIs in Taiwan.

## 2. Methods

### 2.1. Design

A mixed-methods study was conducted to assess the knowledge and technique of psychiatric nurses when administering intramuscular injections of long-acting antipsychotics (LAIs) [28,29]. This study had two phases: (1) a survey of psychiatric nurses to evaluate their knowledge of LAIs and their implementation practices and (2) qualitative research to understand the barriers that nurses face when choosing a VG injection site.

#### 2.1.1. Phase 1: Quantitative Research

##### Participants

For the quantitative phase of the study, participants were selected from three major public psychiatric hospitals in northern, central, and southern Taiwan. These hospitals were part of Taiwan’s core psychiatric care network. According to Roscoe’s (1975) recommendation, this study used a sample size calculation for research [30]. The guideline suggests a sample size of 10 times the number of variables, considering a projected 10% attrition rate. Consequently, the target enrollment for this study was set at 270 nurses. To be eligible, participants had to be registered nurses with at least one year of clinical experience. However, head nurses and other higher-ups who did not provide direct patient care were excluded from the study. Participants were randomly selected using a simple random sampling method with the help of a random number table. Data were collected using self-reported questionnaires from all three psychiatric hospitals in Taiwan.

##### Instrument

The authors developed the questionnaire based on the existing literature [9,15,16,26,31]. It had two sections. The first section asked for demographic information such as gender, age, education level, nursing experience, and experience with psychiatric nursing, as well as asking about their knowledge source for LAIs and how often they administered them. Participants were also asked if they had received training on the techniques of long-acting antipsychotic injection.

The second section evaluated the nurses’ knowledge of LAIs with 13 questions divided into 2 subcategories: LAI injection site and LAI principles and techniques. The questions were answered with either “yes”, “no”, or “do not know”, and one point was awarded for each correct answer. The total score ranged from 0 to 13, with higher scores indicating a higher level of knowledge. The last part of the questionnaire assessed nursing practice with ten questions divided into two subcategories: LAI injection site and LAI principles and techniques. Nurses answered on a scale of “never” to “always”.

The questionnaire was evaluated by a panel of six nursing experts (two academics and four master-level nurses). Each expert ranked each item’s appropriateness, accuracy, and representativeness from 1 to 4 (1 as irrelevant, 2 as requires significant revision, 3 as minor revision needed, and 4 as relevant, clear, and precise). The questionnaire items were revised according to the experts’ suggestions. During the data collection process, a pilot study was conducted with 20 participants to evaluate their understanding and the time needed to complete the study. Based on the results of the pilot study, no changes were considered necessary.

The questionnaire examined LAI knowledge, practice, and administration in psychiatric nurses. The questionnaire’s test–retest reliability was calculated with 15 nurses at 14-day intervals. The intraclass correlation coefficient was 0.81 (*p* < 0.0001). Cronbach’s alpha coefficients were 0.89 and 0.82 for the knowledge and practice items, respectively, indicating that the questionnaire had good content validity [32]. The questionnaire was used to examine LAI knowledge, practice, and administration in psychiatric nurses.

##### Data Collection and Analysis

This study’s lead author worked with the three hospitals, and a research assistant was available to support the analysis. The assistant reached out to each nurse, explained the purpose of this research, and provided instructions on how to fill out the questionnaires. The collected data were analyzed using SPSS 20.0 software. The results included descriptive statistics that described the demographic data, clinical variables, and scores from the questionnaire. The LAI knowledge score was between 1 and 13. The Pearson correlation coefficient was used to see the connection between the nurses’ knowledge and their demographic data. In contrast, an independent *t*-test or ANOVA was used to find differences between different nurse groups based on demographics and knowledge levels.

#### 2.1.2. Phase 2: Qualitative Research

The researchers selected participants who responded “sometimes” to the items on the injection DG site choice frequency in the questionnaire. The author (M.-L.L.) conducted in-depth interviews in Mandarin to understand why psychiatric nurses chose the injection DG sites and the barriers they faced. The interviews lasted between 40 and 60 min and were recorded and transcribed into Chinese text for analysis. The researchers followed Lincoln and Guba’s guidelines for ensuring data analysis credibility, transferability, dependability, and confirmability [33]. The data were analyzed using content analysis by NVivo 12 Pro software. The researchers continued the research until theme saturation and confirmed the results from 20 participants. Finally, the team validated the results using feedback from 10 participants.

##### Ethical Considerations

This study was approved by the Human Experiment and Ethics Committees of the three psychiatric hospitals (IRB-102020). The participants received a full explanation of their rights to anonymity, confidentiality, and freedom to withdraw from the study. Written consent was obtained from each participant. All data were kept anonymous and confidential throughout and after this study.

## 3. Results

This study aimed to assess the knowledge, techniques, and actual practices of psychiatric nurses in Taiwan when administering LAIs. A mixed-methods study was conducted. The results show psychiatric nurses’ knowledge of LAIs, their implementation practices, and why nurses chose injection DG sites.

### 3.1. Phase 1: Quantitative Results

#### 3.1.1. Participant Demographics

Of the 270 nurses recruited, 269 participated in this study, with a response rate of 99.6%. Most participants were female (92.9%), with an average age of 33.4 years (SD = 8.20). Overall, 40.9% of the participants were under 29 years old, and 37.2% were between 30 and 39. The majority of nurses had a bachelor’s degree (66.5%) and an average of 10.7 years of nursing experience (SD = 8.24), with 57.6% having less than ten years of experience in nursing. On average, the sample had 8.45 years (SD = 6.91) of psychiatric nursing experience, with 66.6% having less than ten years of experience. In addition, 40.9% of the participants had a clinical nursing ladder level of N2. The most common sources of knowledge on LAIs were senior nursing colleagues (77.7%), school (64.35%), and hospital continuing education (55.4%). Most participants reported administering LAIs one to two times a month (43.5%) (Table 1).

#### 3.1.2. LAI Administration Knowledge

The results from the questionnaire items assessing nurses’ LAI administration knowledge are presented in Table 2. The average participant score was 9.37 (SD = 1.26) out of 13. The knowledge scores were higher among female and male nurses (*t* = −2.25, *p* < 0.05). Additionally, there was a significant difference in knowledge scores between the age groups (F = 7.18, *p* < 0.05). Scheffe post hoc comparisons found that nurses aged 40 years and older had more knowledge than those aged 29 years and below and 30–39 years. A significant difference in knowledge was also found (F = 5.52, *p* < 0.001) among the nursing experience groups. Scheffe post hoc comparisons indicated that nurses with 15 or more years of nursing experience had more knowledge than those with 5 or fewer years (Table 2).

Finally, positive correlations were found between nurse knowledge and the variables age (*r* = 0.21, *p* < 0.01), nursing work experience (*r* = 2.3, *p* < 0.01), and psychiatric nursing work experience (*r* = 1.6, *p* < 0.01). There was a positive association between nurse age and their overall LAI knowledge (*r* = 0.12, *p* < 0.01).

#### 3.1.3. LAI Administration Practice

The relationships between psychiatric nurse characteristics and LAI administration practice are shown in Table 3. The majority of nurses (N = 149, 55.4%) reported using the two-needle technique (question 1, Table 3). Regarding using the deltoid muscles for the injection site (question 2, Table 3), 84 nurses (31.2%) always chose the deltoid muscles, approximately half (N = 147, 54.6%) sometimes chose the deltoid muscles, and 38 (14.1%) never chose them.

Comparing the use of the DG and VG injection sites (questions 3 and 4, Table 3), the nurses had a low rate of choosing the VG site. Of the nurses, 100 (37.2%) always chose the DG site for LAIs, approximately half (N = 138, 51.3%) sometimes chose the DG site, and 31 (11.5%) never chose the DG site. Only 30 nurses (11.2%) always chose the VG injection site, one-third (N = 83, 30.9%) sometimes chose the VG site, and 156 (58%) never chose it.

Regarding the use of the Z-track method, more than half of the nurses always used it (question 5, Table 3; N = 155, 57.6%), approximately one-fourth (N = 68, 25.3%) sometimes used it, and 46 (17.1%) never used it. Close to half of the participants never used the air-bubble technique (question 5, Table 3; N = 128, 47.6%), one-third sometimes used it (N = 86, 32%), and one-fifth always used it (N = 55, 20.4%). Regarding the direct injection technique (question 7, Table 3), approximately one-third of the nurses always used it (N = 86, 32%), 76 (28.3%) sometimes used it, and 128 never used it.

Most participants chose to hold the needle at a 90° angle to the skin for LAI injection (question 8, Table 3; N = 231, 85.9%), insert two-thirds of the needle into the skin (question 9, Table 3; N = 237, 88.1%), and rotate injection sites (question 10, Table 3; N = 230, 85.5%). All of the statuses for implementing LAIs compared to knowledge were not significant.

### 3.2. Phase 2: Qualitative Results

The questionnaire used in this study included a semi-structured, in-depth interview to gain insight into the thought processes of psychiatric nurses when implementing LAIs. The author spoke with 20 nurses to understand the barriers they faced when choosing injection sites and how they overcame them. Two main themes emerged from the analysis: the gap between nurse knowledge and actual practice and lacking confidence and training to use the VG site.

#### 3.2.1. The Gap between Knowledge and Real Practice

While the participants had knowledge about appropriate injection sites, they often were unable to use this knowledge in practice due to patient preference or hesitation. Participant 1, who had worked with many patients during her 25 years of work experience, said, “*As soon as the patient came, he showed his upper arm. I knew the FGA can’t be injected into the deltoid site, so I tried to persuade him to change the injection site, but he persistently refused*.” She continued by saying, “*At that time, I followed his preference, but I pushed [the medication] very slowly and carefully*”.

Patients frequently challenge nurses by rejecting the suggested injection site. Some patients with schizophrenia have more hesitation about gluteal sites because of their symptoms. Participant 13 reported that “*I knew the IMI technique and learned it before, but some of the patients with schizophrenia did not want to expose their buttocks for injection due to their delusions. If they did not trust me, I could not ask them to agree to use the VG site for injection. But I try to tell them about the benefits of using this site. Some patients let me use the VG injection site after they trust me*”. While many nurses learned proper IMI techniques in school, they may be limited in their choice of injection site due to patient preferences or symptoms preventing them from using certain sites safely.

#### 3.2.2. Lacking Confidence and Training to Use the VG Site

Many nurses chose not to try using the VG site for LAI administration; instead, they relied on their current knowledge and practice when administering injections. Participant 6 (14 years of psychiatric nursing experience) said, “*While I had read about the VG injection site in textbooks, my senior nurses taught me about another injection site, so I opted not to use the VG site*”. Participant 8 (six years of nursing experience) knew about the VG site for IMIs but did not use it, saying “*I learned about IMI injections in school, but no one guided me on how to do it in the clinical setting, so I was afraid to hurt the patient. I just used my familiar injection site for patients*”. In summary, while some of the nurses knew about the VG injection site, they need clinical guidance and practice to increase their confidence in using this site.

## 4. Discussion

This study used a mixed-methods approach, combining both quantitative and qualitative methods, to gather a diverse range of perspectives and insights on the implementation of LAIs.

### 4.1. LAI Administration Knowledge and Practice

This study’s results showed that a nurse’s education level had little impact on their knowledge of LAIs. However, gender, age, nursing work experience, and psychiatric nursing work experience were all factors that correlated with higher LAI administration knowledge. Female, older, and more experienced nurses tended to understand LAIs better than male, younger, and less professional nurses. The only exception was that there was no significant correlation between LAI knowledge and the injection techniques used, except for deep injection and rotating the injection site.

In recent years, there has been a push for IMIs to be administered in the VG site, as it does not have a concentration of essential blood vessels or nerves [10,14]. However, despite the support for the VG site, it is not commonly used by clinical nurses [9,16,17,18]. This study found that it was more than just the level of knowledge that influenced nurses’ implementation of IMIs. While some researchers suggest more education for nurses on LAI injection sites [16,19], this study found that knowledge alone did not directly impact their practice.

Based on the qualitative and quantitative findings of this study, administering LAIs requires not only execution by nurses but also obtaining consent from the recipients (patients). Although nurses understand correct injection techniques, patients may not necessarily agree to VG site injections. Therefore, this study suggests that shared decision-making between nurses and patients is necessary. Establishing strong nurse–patient relationships is essential for patients to accept the appropriate injection sites and treatment [34]. In addition to the nurses’ injection knowledge, nurses should actively involve patients in shared decision-making, thereby enhancing patients’ understanding and improving their adherence to the treatment.

### 4.2. Injection Technique Preference

Using the quantitative data of this study, we found that nurses select injection sites based on their knowledge and patient preference. Psychiatric patients can be fixated on receiving their injections in particular regions of the body, often refusing to accept injections at other sites suggested by nurses. Indeed, our findings were similar with previous study [35], indicating that some patients may refuse to receive injections at the VG site. Even when the nurse explains the advantages of the VG site, the patient may still refuse. Patient preference may influence nurses’ implementation of injection sites. For this reason, approximately half of the nurses in this study reported sometimes choosing the deltoid muscles as the injection site for LAIs, although it is not the preferred site. Despite this possibility, nurses must talk to their patients about LAIs and appropriate injection sites based on safety injections for the patient.

In addition, when compared to males, females gained a higher score in terms of the knowledge of intramuscular injection techniques, and nursing personnel who were aged 40 and above also scored higher than nurses aged 29 and under. Past studies found that patients who receive LAIs may experience sclerosis, muscle granulomata, and medication leakage at the injection site [23]. The literature suggested the use of the Air-bubble technique, the Z-track method, and the combination of the air-bubble technique and the Z-track method as a substitute for direct muscle injection [24]. Results from the present study found that over half of the nursing personnel choose the Z-track method as their preferred method for implementing LAIs (57.6%), while one-third of nurses always use the direct muscle injection method for LAIs (32.0%). It is imperative to further promote the Z-track method or the air-bubble technique among nurses.

### 4.3. Nurses’ Clinical Training

The administration of long-acting injectables (LAIs) in clinical nursing presents two distinct problems: knowledge and implementation. The first aspect, knowledge, is a significant challenge in LAI administration. According to a study, 58% of 269 participants have yet to use the ventrogluteal (VG) site as their preferred injection site. This may be due to their prior training in school, which focused mainly on using the deltoid (DG) site. According to this issue, mental health professional curricula should be revised to include basic clinical skills in locating the VG site on patients. In addition, practicing nurses who require more confidence in administering VG injections need further clinical training to fill this gap.

Despite the need for continuing education in basic nursing techniques, many nurses need more opportunities for professional development after entering the workplace. Clinical nurses typically acquire intramuscular injection (IMI) knowledge from senior colleagues, but due to limited research results, psychiatric nurses may require more knowledge and confidence to practice LAI administration. Although intramuscular injections are an independent nursing technique in clinical practice, some nurses may still need assistance in implementing this technique in real-life situations. To overcome this challenge, they need to understand the problems and discuss their challenges with peers.

Therefore, it is recommended that psychiatric hospitals include clinical courses on LAI techniques as a requirement for continuing education credit. This hands-on experience could enhance patient safety related to LAI administration in these settings.

The second aspect, the implementation aspect, relates to how nurses demonstrate their knowledge of LAIs. In some cases, nurses may know the appropriate VG site and techniques for LAI injections but fail to implement them due to patient refusal. This could be due to the patient’s reluctance to expose their buttocks, which may be linked to their psychotic symptoms. This creates a significant trust issue in psychiatric nursing. Some nurses may wait until the patient trusts them before performing intramuscular injections at the VG site, even if the injection site is inappropriate. This can lead to an ethical dilemma between decency and non-maleficence in nursing. Past research has shown that, even with VG injection training, nurses use fewer VG injections after several years [9,16].

To address these barriers, nurse training should include communication with patients about injection knowledge, starting with changing their perception of LAI injections and reducing their rejection of the VG site. Nurses should arrange positive injection experiences at VG sites with the help of peer support specialists to ease patient worries or symptoms.

In conclusion, the implementation of LAIs in clinical nursing is faced with two challenges: knowledge and implementation. Addressing these challenges requires revisions to health professional curricula, continuing education opportunities, and the inclusion of communication with patients about injection knowledge in nurse training. The ultimate goal is to enhance patient safety and improve the confidence and knowledge of nurses in LAI administration.

## 5. Conclusions and Recommendations

This study highlights the need for more clinical training and experience in LAI administration for psychiatric nurses. This study found that nurses’ knowledge and patient preference play a role in choosing the injection site, and over half of the nurses preferred using the Z-track method. To address the challenges in LAI administration, revisions to health professional curricula, continuing education opportunities, and improved communication with patients are necessary. The ultimate goal is to enhance patient safety and improve the confidence and knowledge of nurses in LAI administration. Clinical courses on LAI techniques should be included in continuing education for psychiatric hospitals to provide hands-on experience. Nurse training should also focus on changing patient perception of LAIs and reducing their rejection of the ventrogluteal site, which is the preferred site. This study stresses the importance of communication between nurses and patients in addressing barriers and improving patient safety.

## Figures and Tables

**Table 1 healthcare-11-01670-t001:** The demographic characteristics of the nurses (N = 269).

Variable	N (%)
Gender	
Male	19 (7.1%)
Female	250 (92.9%)
Age (years), mean = 33.41 (SD = 8.20)	
Under 29	110 (40.9%)
30–39	100 (37.2%)
≥40	59 (21.9%)
Education	
College	69 (25.6%)
University	180 (66.5%)
Graduate	21 (7.8%)
Total years of nursing work, mean = 10.70 (SD = 8.24)	
Fewer than 5 years	84 (31.2%)
5–9.9 years	71 (26.4%)
10–14.9 years	35 (13.0%)
More than 15 years	79 (29.4%)
Years of psychiatric nursing work, mean = 8.45 (SD = 6.91)	
Fewer than 5 years	113 (42.0%)
5–9.9 years	65 (24.2%)
10–14.9 years	33 (12.3%)
More than 15 years	58 (21.6%)
IM injection knowledge (multiple choice)	
From school	173 (64.3%)
From senior colleagues	209 (77.7%)
From self-learning	45 (16.7%)
From drug instruction sheet	91 (33.8%)
From hospital continuing education	149 (55.4%)
Others	121 (45.0%)
Frequency of injecting long-acting antipsychotics	
<1 time per month	66 (24.5%)
1–2 times per month	117 (43.5%)
3–4 times per month	45 (16.7%)
>5 month	41 (15.2%)

**Table 2 healthcare-11-01670-t002:** Participant Knowledge About First-Generation LAIs (N = 269).

Variable	Knowledge of LAI M (SD)	*t*, F, *r* (*p*)
Gender		−2.25 *
Male	7.84 (1.11)	
Female	8.39 (1.01)	
Age (years)		7.18 ***
① ≤29	8.07 (1.05)	
② 30–39	8.51 (1.02)	
③ ≥40	8.59 (0.89)	
	Scheffe ③ > ① ③ > ②	
Total years of nursing work		5.52 **
① <5 years	8.00 (1.12)	
② 5–9.9 years	8.42 (1.00)	
③ 10–14.9 years	8.43 (0.98)	
④ ≥15 years	8.62 (0.88)	
	Scheffe ④ > ①	
Age		0.21 **
Total years of nursing work		2.3 **
Total years of psychiatric nursing work		1.6 **

* *p* < 0.05. ** *p* < 0.01. *** *p* < 0.001.

**Table 3 healthcare-11-01670-t003:** Psychiatric Nursing Preference Practice when Administering First-Generation LAIs (N = 269).

	① Never *n* (%)	② Sometimes *n* (%)	③ Always *n* (%)	*F* (*p*)
I will use one needle to draw medication and another needle for injection (two needles).	46 (17.1)	74 (27.5)	149 (55.4)	0.01
2. I will choose the deltoid injection site when implementing long-acting antipsychotic IM injection.	38 (14.1)	147 (54.6)	84 (31.2)	0.92
3. I will choose the DG site when implementing long-acting antipsychotic IM injection.	31 (11.5)	138 (51.3)	100 (37.2)	0.43
4. I will choose the VG site when implementing long-acting antipsychotic IM injection.	156 (58.0)	83 (30.9)	30 (11.2)	0.42
5. I will use the Z-track method for IM injection.	46 (17.1)	68 (25.3)	155 (57.6)	2.27
6. I will use the air-bubble technique for IM injection.	128 (47.6)	86 (32.0)	55 (20.4)	0.85
7. I will use the direct injection technique for IM injection.	107 (39.8)	76 (28.3)	86 (32.0)	1.49
8. I will hold the needle 90° to the skin when injecting the long-acting antipsychotics.	4 (1.5)	34 (12.6)	231 (85.9)	0.02
9. I will insert two-thirds of the needle into the skin when injecting long-acting antipsychotics.	3 (1.1)	29 (10.8)	237 (88.1)	4.98 ** Scheffe ③ > ②
10. I will rotate different sites for long-acting antipsychotic injection.	3 (1.1)	36 (13.4)	230 (85.5)	4.98 ** Scheffe ③ > ②

** *p* < 0.01.

## Data Availability

All data were generated at the Tsao Psychiatric Center, Taiwan. The derived data supporting the findings of this study are available from the corresponding author Lin, M. L. on request.

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
