# Peer review of "Psychiatric Nurses’ Knowledge and Practice Barriers to Administering Long-Acting Injectable (LAI) Antipsychotics in Taiwan: A Mixed-Methods Study"

_healthcare, 2023, doi:10.3390/healthcare11121670_

Round 1

Reviewer 1 Report

I appreciated this manuscript, even though I identified some concerns about data interpretation and discussion which may improve the significance of results and the appealing to the audience For these reasons, I would recommend the paper for publication with minor revisions. 

If you elect to revise and resubmit the paper, please include a letter which details your changes to the paper or your rationale for not making a suggested change.

In the background section, I suggest the authors highlight that LAIs are advantageous drugs in the care of patients with schizophrenia because, in addition to improving adherence, they are associated with greater subjective well-being and quality of life (doi: 10.1177/2045125315596897). 

Lines 284-289: Share decision model is an important basis of medication adherence. Moreover, I suggest author to the delete “In addition to the nurses' injection knowledge, there is a need to strengthen their ability to persuade patients to accept appropriate injection sites”. It would be preferable to discuss this aspect in terms of the therapeutic relationship. The caregiver and the mental health professional are essential figures for the care of patients and the central point is a good therapeutic relationship. Therefore, nurses should improve the therapeutic relationship in view of a share decision model, the improvement of patients' insight with the goal of improving adherence to therapies (doi: 10.1708/2098.22686).

Author Response

  1. Thank you very much for providing valuable suggestions. Indeed, your advice will significantly enhance the discourse of the manuscript. In the background section, we have incorporated relevant content as suggested, specifically in lines 31-33.
  2. Thanks for your suggestions. We delete “In addition to the nurses' injection knowledge; there is a need to strengthen their ability to persuade patients to accept appropriate injection sites”, and change to describe the relationship between nurses and patients. Please refer to the line 304-308.

Reviewer 2 Report

The manuscript is well prepared.  The reviewer has the following comments to make:

1.The background of the study consists of the majority of citations being not directly linked to long-acting antipsychotics. The reviewer is aware that the literature is less, but the search reveals sufficient enough to include them in the background of the study to strongly justify the need for this study. Hence, modification in the background is suggested. (This comment is made on the observation that in the background, out of 27 citations, 13 are linked to long-acting antipsychotics).

2. On page 2, line 80 mentions "has increased from 29% in 2006 to 9% in 2012[9, 16]", there appears to be some error in the percentage mentioned.

Author Response

  1. I appreciate the reviewer for their valuable suggestions. However, since this study specifically examines the injection knowledge and skills of psychiatric nurses regarding long-acting injections, it is essential to provide the historical background and relevant knowledge of such injections in the introduction. As a result, there is a relatively higher number of literature citations in this study.
  2. Thank you for bringing this to my attention. It is indeed necessary to make the appropriate revisions. We will provide a detailed description of the literature content and make the necessary modifications on the second page, specifically in lines 86-93.

Reviewer 3 Report

Very interesting topic, the researchers did a great job to examine the long-acting injectable antipsychotics knowledge, practice, and administrative barriers in psychiatric nurses from three public psychiatric hospitals in Taiwan. However, the article could be strengthened through:

Introduce every acronym before using it in the text. The first time you use the term, put the acronym in parentheses after the full term, also double check LAI, “Long-acting injectable (LAI) antipsychotics”.

Add more towards scope of the problem in introduction section.

Have you conducted a pilot study for the developed questionnaire?

Elaborate on how you determine the sample size.

Add more current references from the literature.

Good luck

Author Response

  1. We agree with you. We acknowledge your importance regarding including the entire term before introducing the acronym. After a thorough review of the manuscript, we have ensured that the first occurrence of the abbreviation is accompanied by its corresponding full term.
  2. The scope of the research questions in the introduction section has been adjusted as suggested.
  3. We appreciate your suggestion. Due to constraints on word count, we previously omitted this paragraph. However, we have now incorporated the details of the pilot study process into lines 146-149.
  4. The sample size calculation method has been added to lines 122-125.
  5. We have updated the references section to include five recent publications. Please refer to the references for further details.
